# Performance Evaluation of Microscanner Plus, an Automated Image-Based Cell Counter, for Counting CD4+ T Lymphocytes in HIV Patients

**DOI:** 10.3390/diagnostics14010073

**Published:** 2023-12-28

**Authors:** Woong Sik Jang, Junmin Lee, Seoyeon Park, Chae Seung Lim, Jeeyong Kim

**Affiliations:** 1Departments of Emergency Medicine, College of Medicine, Korea University Guro Hospital, Seoul 08308, Republic of Korea; plasmid18@korea.ac.kr; 2Departments of Laboratory Medicine, College of Medicine, Korea University, Seoul 02841, Republic of Korea; dlwnsals15@korea.ac.kr (J.L.); malarim@korea.ac.kr (C.S.L.); 3Departments of Laboratory Medicine, College of Medicine, Korea University Guro Hospital, Seoul 08308, Republic of Korea; 08tjdus@naver.com

**Keywords:** CD4, cell count, flow cytometry, Microscanner, HIV

## Abstract

Counting CD4+ T lymphocytes using flow cytometry is a standard method for monitoring patients with HIV infections. Simpler and cheaper alternatives to flow cytometry are in high demand because getting access to flow cytometers is difficult or impossible in resource-limited settings. We evaluated the performance of the Microscanner Plus, a simple and automated image-based cell counter, in determining CD4 counts against a flow cytometer. CD4 count results of the Microscanner Plus and flow cytometer were compared using samples from 47 HIV-infected patients and 87 healthy individuals. All CV% for precision and reproducibility tests were less than 10%. The Microscanner Plus’s lowest detectable CD4 count was determined to be 15.27 cells/µL of whole blood samples. The correlation coefficient (R) between Microscanner Plus and flow cytometry for CD4 counting in 134 clinical samples was very high, at 0.9906 (*p* < 0.0001). The automated Microscanner Plus showed acceptable analytical performance for counting CD4+ T lymphocytes and may be particularly useful for monitoring HIV patients in resource-limited settings.

## 1. Introduction

Human immunodeficiency virus (HIV) infections were responsible for over six hundred thousand deaths worldwide in 2022, according to the World Health Organization (WHO) [1]. Approximately thirty-eight million people have been diagnosed with HIV, and an estimated 1.7 million new cases are reported each year. The continuing increase in HIV infections despite efforts by the WHO, especially in developing countries, is the most critical public health concern [2]. Recognition of people with HIV infections is difficult due to the long-term asymptomatic initial stages of infection and the various routes of transmission. According to the Centers for Disease Control and Prevention (CDC), up to 14.2% of patients are unaware of their infections, which represents a significant proportion of up to 38% of all reported cases [3]. The existence of a clinical latency phase, during which about 50% of untreated patients develop AIDS over the course of 10 years, remains an obstacle to initiating timely treatment and preventing transmission [4]. Characteristics of high transmission rates, variable latency periods, progression to AIDS, and high rates of mortality represent major public health challenges in many developing countries [5,6].

While mortality rates and comorbidities associated with HIV infections have declined, complete eradication or cure remains almost impossible. Many people living with HIV continue to present with low CD4+ T lymphocyte counts (CD4 counts), categorized as advanced disease [7]. Nonetheless, early initiation of highly active antiretroviral therapy (HAART) prior to the onset of immunosuppression enables similar life expectancies to non-infected individuals [8]. Prompt initiation of HAART before the development of AIDS is therefore the most important factor in reducing mortality and complication rates [9,10,11]. Although complete viral eradication and HIV cure are not yet available, if HAART is started in a timely manner before immunodeficiency develops, life expectancy may be similar to that of people living without HIV [8]. Initiation of HAART requires a comprehensive evaluation of the stages of HIV infection, ranging from primary infection to clinical stages 1–4, which are characterized by variable symptoms and signs [12].

Tests available for monitoring people living with HIV include HIV RNA, CD4+ T lymphocyte count, and complete blood count (CBC), of which the CD4+ T lymphocyte count is the most frequently used indicator in the evaluation of HIV infection [13,14,15]. CD4+ T lymphocyte count exhibits a significant correlation with HIV RNA count in the context of HAART treatment response and is a reliable marker of the potency of the immune system [16]. It has been reported that 65% of newly diagnosed cases of HIV infections in South Korea exhibit CD4+ T lymphocyte counts of 350 cells/μL or higher [1]. This is notably higher than the range of 20~30% observed in other developing countries, suggesting relatively early detection of infection while the immune function is well-maintained and the risk of progression to AIDS is low. Current treatment guidelines recommend that patients begin treatment as soon as possible once they are confirmed to be infected with HIV. However, the rating of evidence supporting this recommendation is more concrete (AIa, Strength: Strong panel support, Quality: Evidence from 1 or more randomized controlled trials published in peer-reviewed journals) for those with CD4+ T lymphocyte counts below 500 cells/μL, whereas the level of evidence is lower (BIII, Moderate panel support, based on panel’s analysis of available evidence) when CD4+ T lymphocyte counts exceed 500 cells/μL [17,18]. Therefore, the CD4+ T lymphocyte count serves as the most important and reliable marker for staging HIV infections.

Flow cytometry is the current gold standard suggested by the WHO for counting CD4+ T lymphocytes; however, more than 3 million people living with HIV reside in low-to-middle-income countries where access to flow cytometry analyzers is limited, and it is in these developing countries where HIV infections are more prevalent. Requirements for skilled operators, costs, and resource-dependent characteristics of flow cytometry result in limited access in such resource-limited settings [19,20]. The recommendation made by the WHO to initiate ART regardless of CD4+ T lymphocyte counts in diseases at clinical stages 3 and 4 is attributed to the limited access to CD4+ T lymphocyte counting [21].

The Microscanner Plus (Biozentech, Seoul, Republic of Korea) is a cell counter that is smaller and cheaper than a flow cytometer and can be used by a non-expert. In this study, we utilized the Microscanner Plus to measure CD4 counts in patients with HIV infections. To validate the performance of the Microscanner Plus, a comparative evaluation was performed using flow cytometry as a standard method of CD4+ T lymphocyte counting. The analytical performance of Microscanner Plus was verified through precision, reproducibility, and correlation testing. The clinical usefulness of Microscanner Plus for measuring CD4 counts in patients with HIV infections in resource-limited settings was studied through analytical and clinical performance evaluation.

## 2. Materials and Methods

### 2.1. Sample Collection and Study Design

A total of 47 people living with HIV and 87 people living without HIV who visited Korea University Guro Hospital from September 2019 to February 2023 participated in this study. All samples were collected as ethylenediaminetetraacetic acid (EDTA) anticoagulated venous blood specimens and processed immediately. Using 134 clinical samples, the performance of the Microscanner Plus was compared to that of the FACSLyric flow cytometer (Beckton Dickinson, Franklin Lakes, NJ, USA). The protocols used in this study were implemented in accordance with the Helsinki Declaration and were reviewed and approved by the institutional review board (IRB No. 2019GR0458).

### 2.2. Flow Cytometry

The FACSLyric was used as a flow cytometer for counting CD4 in people living without HIV and in people living with HIV. A tube containing a mixture of 200 µL of peripheral blood, 5 µL of BB515-CD45 (BD Biosciences, Franklin Lakes, NJ, USA), and 5 µL of PE-CD4 (BD Biosciences, NJ, USA) was incubated at room temperature in the dark for 15 min. After staining the cells, the mixture was added to 1 mL of Lysing solution OptiLyse C (BD Biosciences, NJ, USA) and then incubated for 10 min. The cells were then centrifuged at 2000× *g* rpm for 5 min after adding 5 mL of PBS (Phosphate-buffered saline, pH 7.4). The supernatant was removed and the cells resuspended in 1 mL PBS. The resuspended sample (1 mL) was then loaded into the flow cytometer and the CD4/CD45 ratio calculated using flow cytometry. The CD4 gating strategy was performed with reference to a previous study [22].

### 2.3. Microscanner Plus

The Microscanner Plus is a bench-top-sized instrument with 14.4× magnification capability. A bright field (BF) image is obtained using a built-in CMOS sensor camera and a separate fluorescence green light source at 540 nm, with no emission filter. All power LEDs are the same at 1400 mA. The green field (GF) image uses a blue light source at 450 nm, with an emission filter (530 ± 25 nm). The red field (RF) uses a green light source at 530 nm, with an emission filter (605 ± 27.5 nm). The chip used in this study is specified as BZ-3, with dimensions of 25 × 75 × 1.7 mm (W × D × H). The depth of the channel is 100 µm and the amount of sample inserted is 45 µL. Briefly, the analytical process is as follows: A fluorescence-labeled sample is first added to the BZ-3 microchip (Biozentech, Seoul, Republic of Korea) (Figure 1A,B), which is then loaded onto the Microscanner Plus (Figure 1C). After the instrument automatically photographs and analyzes the sections of the microchip with integrated auto-focusing, the built-in imaging analysis software for the Microscanner Plus implements a clustering algorithm. The automated clustering program enumerates the cells present within the image and generates a dot plot of percentages. The Microscanner Plus represents the green field intensity in CH1 and the red field intensity in CH2 (Figure 1D). The clustering can be conducted for both CH1 (green field intensity) and CH2 (red field intensity), for which a histogram representing the intensity is generated (Figure 1E). The Microscanner Plus is capable of processing between 7 and 10 samples per hour.

A tube containing a mixture of 200 µL of peripheral blood, 10 µL of BB515-CD45 (BD Biosciences, Franklin Lakes, NJ, USA), and 10 µL of PE-CD4 (BD Biosciences, NJ, USA) was incubated at room temperature in the dark for 15 min. After staining the cells, the mixture was added to 1 mL of Lysing solution OptiLyse C (BD Biosciences, NJ, USA) and incubated for 10 min. The cells were then centrifuged at 2000× *g* rpm for 5 min after adding 5 mL of PBS (Phosphate-buffered saline, pH 7.4). The supernatant was removed and the cells resuspended in 1 mL PBS. The resuspended sample (45 µL) was loaded onto a BZ-3 microchip, which was then inserted into the Microscanner Plus. The CD4/CD45 ratio was calculated using the Microscanner Plus image analysis program.

### 2.4. Counting CD4+ T Lymphocytes

The CD4+ T lymphocyte count was calculated using the following equation:CD4 count (cells/µL) = CD4/CD45% × total WBC numbers (cells/µL)

The WBC number of each blood sample was measured using the Automated Hematology Analyzer DxH900 (Beckman Coulter, Brea, CA, USA). CD4 and CD45 fractions of WBC were measured using both the Microscanner Plus and the FACSLyric flow cytometer. Based on this measured value, the number of CD4 + T lymphocytes can be calculated by dividing the CD4 fraction by the CD45 fraction and then multiplying by the WBC count.

### 2.5. Evaluation of the Performance of Microscanner Plus

The analytical performance of the Microscanner Plus for determining CD4 counts was evaluated based on precision and reproducibility using the coefficient of variation (CV). For precision analysis, the CV value was obtained from the standard deviation (SD) divided by the mean of five repeat results of three samples analyzed every 5 days. For reproducibility, the CD4 count of each sample was analyzed by four independent researchers using the same three samples. Based on the CD4 counts analyzed by four independent experimenters, the CV value of the CD4 count between experimenters was calculated to confirm reproducibility. ANOVA test was used to verify statistical differences in CD4 counts obtained by the four independent experimenters. To determine the detection limit of CD4 count using Microscanner Plus, 7 diluted samples were fluorescently stained and measured 5 times for each concentration. In this study, the lower limit of detection was determined as the lowest concentration in the sample that satisfied a CV of less than 5% for CD4 count.

The clinical performance of the Microscanner Plus for CD4 count measurement was assessed by comparing it with that of the flow cytometer in a total of 134 samples from 47 HIV-infected patients and 87 healthy individuals. Correlation, regression, and Bland-Altman analyses were conducted.

### 2.6. Statistical Analysis

Correlation, regression, and Bland-Altman analyses were conducted. A *p*-value of less than 0.01 was considered statistically significant. Microsoft Excel 2019 (Microsoft, Redmond, WA, USA), GraphPad Prism software version 10.1.0 (GraphPad Software, La Jolla, CA, USA), MedCalc version 22.016 (MedCalc software Ltd., Ostend, Belgium), and Labostats software version 1.5.3.0 (Laboratory Medicine Foundation, Seoul, Republic of Korea) were used for statistical analysis in this study.

## 3. Results

### 3.1. Precision

The precision of the Microscanner Plus for determining CD4 counts in three samples with different concentrations (1000 cells/μL, 500 cells/μL, and 170 cells/μL) was tested in five replicate tests every five days. The CD4 count was obtained by multiplying the CD4/CD45 ratio (%) measured using the Microscanner Plus with the total white blood cell count measured using the DxH900 automated hematology analyzer. Mean CD4 counts at the three levels were 999.40, 498.78, and 168.15, respectively. The repeatability at the three levels was 3.3%, 6.6%, and 2.4%, respectively. The three samples showed average CD4 counts of 781.91, 490.30, and 166.64 cells/µL, respectively. The SDs of the three samples were 42.70, 38.45, and 4.39, respectively (Table 1). When the above results were analyzed using Labostats software, repeatability was 3.3%, 6.6%, and 2.4%, respectively, and total imprecisions were 3.1%, 6.3%, and 2.2%, respectively.

### 3.2. Reproducibility

In the reproducibility test, four experimenters—A, B, C, and D—performed measurements of CD4 counts. Experimenter A’s measurements yielded mean CD4 counts of 1499.40, 740.17, and 314.57 cells/µL for the respective samples, with corresponding standard deviations (SDs) of 11.90, 4.75, and 2.46 cells/µL. The coefficient of variations (CV) for these measurements was maintained below 1%, specifically at 0.79%, 0.64%, and 0.78%. Similarly, experimenter B reported mean CD4 counts of 1487.50, 746.13, and 314.20 cells/µL, with SDs of 20.61, 3.55, and 2.08 cells/µL, and CVs of 1.39%, 0.48%, and 0.66%. Experimenter C’s results had mean values of 1483.53, 741.77, and 317.73 cells/µL, with SDs of 24.77, 6.87, and 3.55 cells/µL, and CVs of 1.67%, 0.93%, and 1.12%. Lastly, experimenter D recorded means of 1495.43, 741.77, and 309.80 cells/µL, SDs of 13.74, 6.87, and 8.75 cells/µL, and CVs of 0.92%, 0.93%, and 2.82% (Table 2). All intra-individual CV values were within 5%, showing excellent reproducibility. Additionally, the ANOVA test revealed no statistically significant differences in CD4 counts among the four independent experimenters, as indicated by an F-statistic of 1.98 × 10^−5^ and a *p*-value of 1.0.

### 3.3. Limit of Detection

The CVs of the samples were 4.21%, 4.64%, 4.89%, 2.35%, 4.73%, 3.09%, and 6.84%, respectively. Because the lower limit of detection was determined to be the lowest concentration in the sample that had a CV of less than 5% for CD4 count, the LOD of the Microscanner Plus was determined to be 15.27 cells per 1 μL of whole blood sample (Table 3).

### 3.4. Clinical Performance of the Microscanner Plus

Analysis of CD4 counts, calculated using the CD4/CD45 ratio and WBC count, in a total of 134 patients exhibited a remarkably high correlation between Microscanner Plus and flow cytometry, with an R^2^ value of 0.9906. The CD4/CD45 ratio also showed a high correlation (R^2^ = 0.9694) between the Microscanner Plus and flow cytometry methods (Figure 2). Additionally, Bland-Altman analysis was performed to visually demonstrate the correlation between CD4 counts obtained using the Microscanner Plus and flow cytometry (Figure 3). These results suggest a consistent and reliable relationship between the CD4 counts obtained using the two methods.

## 4. Discussion

Flow cytometry is the gold standard for measuring CD4+ T lymphocytes; however, it is costly and labor intensive, and the assay is complex, requiring a specialist. To overcome the limitations of flow cytometry, imaging microscopy that automatically quantifies the biological properties of images is becoming more widespread [23,24]. Many devices that can act as alternatives for flow cytometry have been developed. Recently, FACSCount, Guava EasyCD4 Volumetric System, and an Image-Based CD4+ T Lymphocyte Counting System have been developed and introduced as cost-effective alternatives for expensive flow cytometry. Among these, the automatic image analyzer analyzes images of cells or pathogens in a sample through simple manipulation and generates images from various specimens or stained samples through optical and two fluorescence channels and obtains the information necessary for image analysis, such as the cell count and concentration [25,26,27,28,29]. Cytiva’s Cell Analyzer 6000 (GE Healthcare Bio-Sciences Corp., Marlborough, MA, USA) can image specimens at optimized speed and quality using a sensitive laser-based confocal imaging platform. Other automated cell counters such as Countess (Invitrogen, Waltham, MA, USA), TC10 (BioRAD, Hercules, CA, USA), MINI (Nexcelom, Lawrence, MA, USA), and Mozi (ORFLO, Ketchum, ID, USA) were recently released [30]. Common disadvantages of the current methods include the complexity of the interface and the need for independent analysis software. Furthermore, it must be considered that most automated cell counters are limited in focus to only observing mammalian cells [31].

In this study, we evaluated the performance of Microscanner Plus, an automated imaging cytometer, as a potential alternative to flow cytometry for measuring CD4 counts. The rationale for exploring alternatives is the limited access to flow cytometry in resource-limited settings, where a significant number of HIV-infected patients reside. In the precision analysis of the Microscanner Plus, repeatability was 3.3%, 6.6%, and 2.4%, and total imprecisions were 3.1%, 6.3%, and 2.2%, respectively. Therefore, all CVs in the precision analysis were less than 10%, indicating sufficient performance for clinical use. The reproducibility of the Microscanner Plus was determined to be less than 5% CV of the CD4 count when repeated three times by four experimenters. There were no statistical differences in CD4 counts between the four independent experimenters based on the ANOVA test. In addition, the LOD of the Microscanner Plus was found to be 15.27 cells/μL of whole blood samples, which is believed to be sufficient for monitoring CD4 counts in HIV patients. Finally, when comparing the CD4/CD45 ratio and the CD4 count values obtained using the Microscanner Plus and flow cytometry for 134 clinical samples, including those with HIV, the correlation coefficients (R^2^) were 0.9694 and 0.9906, respectively. Analysis of the correlation between the Microscanner Plus and flow cytometer results revealed a high level of agreement, both in artificially diluted samples and in clinical samples. The Bland-Altman plot reinforced the strong correlation, emphasizing the consistency and accuracy of the Microscanner Plus in measuring CD4 counts. These results demonstrate that the Microscanner Plus shows a high correlation with flow cytometry for CD4 count measurements and exhibits high precision, reproducibility, and consistency, regardless of the user’s expertise or experience. Additionally, our study included a comparison of CD4 count results obtained from the FACSLyric and the Microscanner Plus. This comparison focused on 46 samples, each characterized by a lymphocyte/WBC ratio of less than 20%. The results demonstrated a high correlation between these two methods, as evidenced by an R^2^ value of 0.992 (Appendix A), indicating a very strong agreement in quantifying CD4 counts in lymphopenic samples. Furthermore, we investigated the correlation between CD4 counts obtained using the Microscanner Plus and those obtained using the gold standard method of FACS analysis with lymphocyte gating. This exploration also revealed a high degree of correlation, with an R² value of 0.9833 (Appendix A). Therefore, these findings suggest that the Microscanner Plus instrument can serve as an excellent alternative to flow cytometry for CD4 count, particularly in resource-limited settings.

In this study, immunophenotyping using Microscanner Plus was performed in HIV patients to calculate the patients’ CD4 cell counts, and the analytical performance was considered acceptable for clinical use. This study addresses the limitations of current methods, especially in resource-limited settings, and proposes the Microscanner Plus as a cost-effective and accessible alternative. The use of an automated imaging cytometer aligns with the need for simpler and more widely applicable technologies in regions with limited resources. Indeed, while the FACS Lyric is priced between $100,000 and $500,000, the Microscanner Plus is comparatively more affordable at around $20,000. This cost difference makes it possible to elevate the applicability of the Microscanner Plus in resource-limited laboratory settings, especially in regions with a high prevalence of HIV-infected patients. Further research and validation in diverse clinical settings would be beneficial to solidify the potential role of Microscanner Plus in improving HIV patient care globally.

In addition, the Microscanner Plus is expected to be used in a variety of clinical fields if its analysis performance is verified using more antibodies or other clinical samples. For example, Microscanner Plus has the potential to be used for immunophenotyping in other diseases such as blood cancers or innate immune disorders.

## 5. Conclusions

Microscanner Plus exhibited acceptable analytical performance for immunophenotyping assays measuring CD4 counts in HIV patients. This study provides robust evidence supporting the effectiveness of the Microscanner Plus in measuring CD4 counts in HIV-infected patients. Furthermore, the Microscanner Plus offers the advantage of significantly lower cost and easier operation compared with conventional flow cytometry methods. Therefore, the Microscanner Plus is expected to replace flow cytometry in resource-limited settings for counting CD4+ T lymphocytes in HIV patients.

## Figures and Tables

**Figure 1 diagnostics-14-00073-f001:**
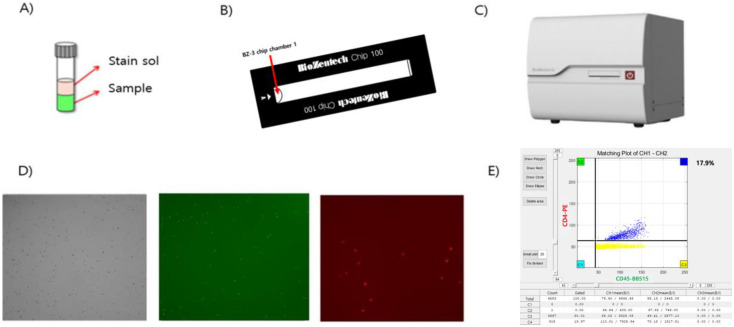
Schematic diagram of the microchip and the Microscanner Plus equipment. A cell-counting program based on image processing. After injecting the stained sample (**A**) onto the microchip (**B**), the prepared chip is placed in the Microscanner Plus (**C**). When the sample is injected into the microchip hole, it spreads evenly after about 10 s. Real fluorescent image (**D**) of the microchip and scatter plots (**E**) of CD4/CD45 are generated using the clustering program (Biozentech, Seoul, Republic of Korea).

**Figure 2 diagnostics-14-00073-f002:**
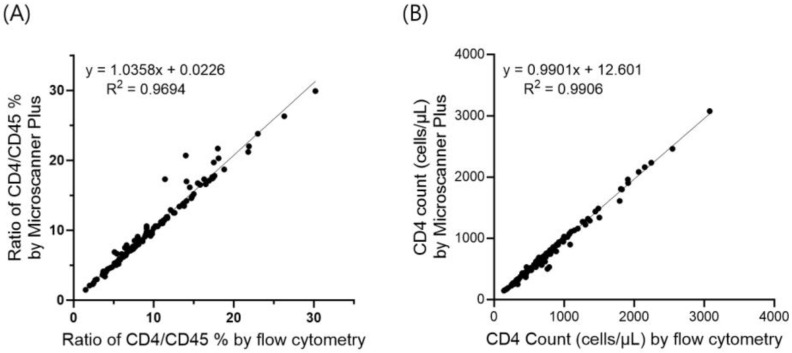
Correlation between CD4/CD45 ratio (**A**) and CD4 counts (**B**) measured using Microscanner Plus (MSP) and Flow cytometry (FCM) in 134 clinical samples. Black dots indicate the CD4/CD45% results in each clinical sample.

**Figure 3 diagnostics-14-00073-f003:**
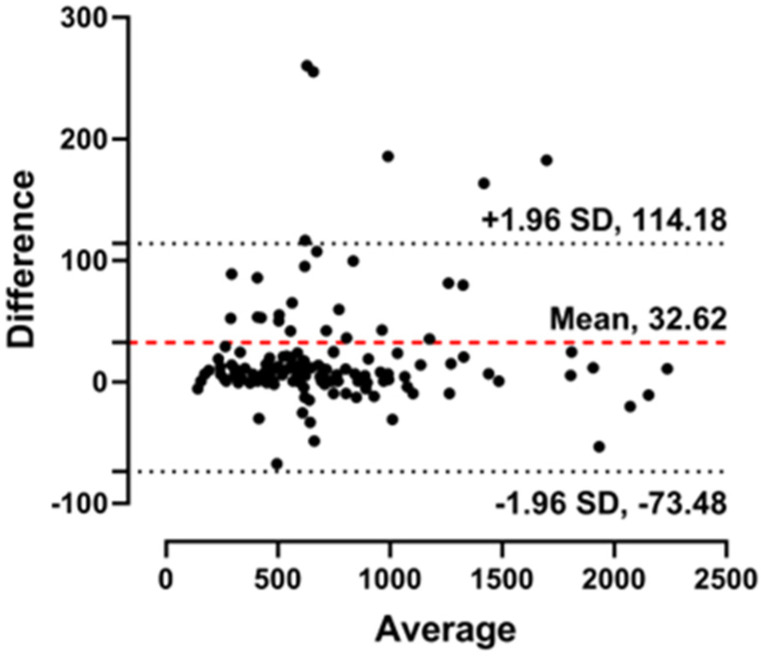
A Bland-Altman plot to detect differences in CD4 counts measured using Flow cytometry and Microscanner Plus equipment (N = 134). The mean difference (Red dashed line) and limit of agreement (Black dashed lines) are plotted. The solid line indicates the 95% limits of agreement (Mean difference ± 1.96 SD). Black dots indicate the CD4 counts results in each clinical sample.

**Table 1 diagnostics-14-00073-t001:** Measurement results of CD4 counts showing the precision of the Microscanner Plus.

Sample	Mean (Cells/µL)	SD (Cells/µL)	Repeatability (CV%)	Total Imprecision (CV%)
1	999.40	31.52	3.3	3.1
2	498.78	19.93	6.6	6.3
3	168.15	3.77	2.4	2.2

**Table 2 diagnostics-14-00073-t002:** Measurement result of CD4 counts showing the reproducibility of the Microscanner Plus.

Sample	Tester	Mean (Cells/µL)	SD (Cells/µL)	CV (%)
1	A	1499.40	11.90	0.79
	B	1487.50	20.61	1.39
	C	1483.53	24.77	1.67
	D	1495.43	13.74	0.92
2	A	740.17	4.75	0.64
	B	746.13	3.55	0.48
	C	741.77	6.87	0.93
	D	741.77	6.87	0.93
3	A	314.57	2.46	0.78
	B	314.20	2.08	0.66
	C	317.73	3.55	1.12
	D	309.80	8.75	2.82

**Table 3 diagnostics-14-00073-t003:** Limit of Detection for CD4 counts using the Microscanner Plus tested in three samples with different concentrations.

Sample	Mean (Cells/µL)	SD (Cells/µL)	CV (%)
1	1002.52	42.16	4.21
2	746.52	34.67	4.64
3	488.10	24.06	4.89
4	166.62	3.91	2.35
5	82.37	3.90	4.73
6	15.27	0.47	3.09
7	6.20	0.42	6.84

## Data Availability

The authors declare that all related data are available from the corresponding author upon reasonable request. The data are not publicly available due to privacy.

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
