# Peer review of "Performance Evaluation of Microscanner Plus, an Automated Image-Based Cell Counter, for Counting CD4+ T Lymphocytes in HIV Patients"

_diagnostics, 2023, doi:10.3390/diagnostics14010073_

Round 1
Reviewer 1 Report
Comments and Suggestions for Authors
general comments:
did the authors try the staining also with the anti-CD3 marker? The authors use 20ul/Abs to stain wb, did they perform a titration of the antibodies? Have they been able to use smaller concentrations of them? How many samples the microscanner works in one hour?How many samples may load into the single cartridge?
The authors explain that the sample is resuspended in 2ml of buffer after the lysis and 45ul of cells are put inside the cartridge. Why the authors resuspend PBMC of 400ul of WB in 2ml of buffer? Have any tests been done with more concentrated cells?
Did they try to analyze lymphopenic samples or with CD45 negative cells?
If there were any blood count reader in laboratories with limited possibilities, did the authors think how they calculate the CD4 T cells count?
The FACS Lyric is an expensive cytometer and also probably the microscanner plus. Did the authors think about the costs for laboratories with limited settings?
Probably the authors should to increase the number of enrolled subjects of the study.
Please, refer to HIV negative and HIV positive individuals as people living without HIV (PLWoH) and people living with HIV (PLWH) in all manuscript.
Introduction:
lines 82-83: cheaper and smaller than what?
Materials and Methods
Lines: 144-145: it is not clear how the authors calculate the CD4 T cells count.
Comments on the Quality of English LanguageMinor editing of the English language
Author Response
Comments and Suggestions for Authors
general comments:
Comments 1: did the authors try the staining also with the anti-CD3 marker?
- Although we did not employ the anti-CD3 marker, when comparing the CD4 counts obtained through lymphocyte gating in FACS analysis with the results of Microscanner Plus, we observed a similarly high correlation, with an R² value of 0.9833. Thus, we added it in discussion section as follows: (Line 307-312)
”Furthermore, we investigated the correlation between CD4 counts obtained from the Microscanner Plus and those derived from the gold standard method of FACS analysis with lymphocyte gating. This exploration also revealed a high degree of correlation, with an R² value of 0.9833 (sFigure 2). Therefore, these findings suggest that the Microscanner Plus instrument can serve as an excellent alternative to flow cytometry for CD4 count, par-ticularly in resource-limited settings.”
Comments 2: The authors use 20ul/Abs to stain wb, did they perform a titration of the antibodies? Have they been able to use smaller concentrations of them?
- Next, there is a mistake in writing the sample preparation method. Instead of using 20 µl of antibody per 400 µl sample, we actually used 10 µl of antibody for 200 µl of whole blood. We discovered that using 5 µl antibody, similar to the amount used in FACS, resulted in decreased sensitivity in Microscanner Plus. We have revised the methods for the Microscanner Plus as follows: (Line 146-159)
“Tube containing a mixture of 200 µl of peripheral blood and 10 µl of BB515-CD45 (BD Bio-sciences, NJ, USA) and 10 µl of PE-CD4 (BD Biosciences, NJ, USA) was incubated at room temperature in the dark for 15 minutes. After staining of the cells, the mixture is pooled in-to 1 ml of Lysing solution OptiLyse C (BD Biosciences, NJ, USA) for 10 minutes and the cells are centrifuged at 2,000 rpm for 5 minutes after adding 5 ml PBS (Phos-phate-buffered saline, pH 7.4). The supernatants are removed, and the cells are resus-pended in 1 ml PBS. After loading the resuspended sample (45 µl) into a BZ-3 microchip, which was then inserted into the Microscanner Plus, the CD4/CD45 ratio was calculated by Microscanner Plus image analysis program.”
Comments 3: How many samples the microscanner works in one hour?
- Regarding the operational capacity of the Microscanner Plus, it can process between 7 to 10 samples per hour. we added it in Materials and Methods section as follows: (Line 137-138)
“The Microscanner Plus is capable of processing between 7 to 10 samples per hour.”
Comments 4: How many samples may load into the single cartridge?
- As for the loading capacity, the BZ-3 cartridge utilized in our study is designed to accommodate one sample. However, an inaccuracy was noted in Figure 1B, where a BZ-1 chip with two chambers was depicted instead. Therefore, we have updated Figure 1B to accurately represent the BZ-3 cartridge.
Comments 5: The authors explain that the sample is resuspended in 2ml of buffer after the lysis and 45ul of cells are put inside the cartridge. Why the authors resuspend PBMC of 400ul of WB in 2ml of buffer? Have any tests been done with more concentrated cells?
- We have identified an error in the description of the sample preparation method. Accordingly, we have revised the methods for both Flow cytometry and the Microscanner Plus as follows: (Line 146-159)
“Tube containing a mixture of 200 µl of peripheral blood and 10 µl of BB515-CD45 (BD Bio-sciences, NJ, USA) and 10 µl of PE-CD4 (BD Biosciences, NJ, USA) was incubated at room temperature in the dark for 15 minutes. After staining of the cells, the mixture is pooled in-to 1 ml of Lysing solution OptiLyse C (BD Biosciences, NJ, USA) for 10 minutes and the cells are centrifuged at 2,000 rpm for 5 minutes after adding 5 ml PBS (Phos-phate-buffered saline, pH 7.4). The supernatants are removed, and the cells are resus-pended in 1 ml PBS. After loading the resuspended sample (45 µl) into a BZ-3 microchip, which was then inserted into the Microscanner Plus, the CD4/CD45 ratio was calculated by Microscanner Plus image analysis program.”
Comments 6: Did they try to analyze lymphopenic samples or with CD45 negative cells?
- As your suggestion, we did include an analysis of CD4 counting of the FACS Lyric and the Microscanner Plus for 46 clinical samples with a low lymphocyte/WBC ratio of less than 20%. The results were quite promising, demonstrating a high correlation between the two methods. The correlation coefficient (R² value) was 0.992 when comparing CD4 counts obtained from CD4/CD45 ratio of the FACS Lyric and the Microscanner Plus. This indicates a very strong agreement between the two methods in quantifying CD4 counts in lymphopenic samples. We have added this content to the Discussion section as follows: (Line 302-307)
“Additionally, our study included a comparison of CD4 count results obtained from the FACS Lyric and the Microscanner Plus. This comparison focused on 46 samples, each characterized by a lymphocyte/WBC ratio of less than 20%. The results demonstrated a high correlation between these two methods, as evidenced by an R² value of 0.992 (sFigure 1), indicating a very strong agreement in quantifying CD4 counts in lymphopenic samples.”
Comments 7: If there were any blood count reader in laboratories with limited possibilities, did the authors think how they calculate the CD4 T cells count?
- If a laboratory has a blood count reader but lacks access to FACS or the Microscanner, one approach is to estimate CD4 counts using absolute lymphocyte counts. This method can be particularly useful in settings where flow cytometry (FACS) or other specialized equipment is not available. Such alternative methods enable the effective measurement of CD4 T cell counts in environments where standard equipment may be inaccessible (Young B, Ng OT, Lye DC, Leo YS. Derivation and validation of an accurate estimation of CD4 counts from the absolute lymphocyte count in virologically suppressed and immunologically reconstituted HIV infected adults. BMC Infect Dis. 2015 Aug 13;15:330. doi: 10.1186/s12879-015-1079-5.).
Comments 8: The FACS Lyric is an expensive cytometer and also probably the microscanner plus. Did the authors think about the costs for laboratories with limited settings?
- As your comment, we considered the cost for advanced cytometry equipment in resource-limited settings in this study. The FACS Lyric is priced between $100,000 to $500,000, while the Microscanner Plus is comparatively more affordable at around $20,000. Our study aimed to provide a cost-effective alternative for such environments, and our findings suggest that the Microscanner Plus offers a reliable yet more accessible solution for CD4 T cell counting. We have added this content to the Discussion section as follows: (Line 319-322)
“Indeed, while the FACS Lyric is priced between $100,000 to $500,000, the Microscanner Plus is comparatively more affordable at around $20,000. This cost difference is possible to elevate the applicability of the Microscanner Plus in resource-limited laboratory settings, especially in regions with a high prevalence of HIV-infected patients.”
Comments 9: Probably the authors should to increase the number of enrolled subjects of the study.
- Following your recommendation, we have expanded our study to include more participants. Specifically, we have added 10 additional HIV-positive patients and 50 HIV-negative individuals. With this expansion, the total number of 134 participants in our study now included 47 HIV-positive patients and 87 HIV-negative individuals. We have updated the related contents throughout the entire manuscript to reflect these changes.
Comments 10: Please, refer to HIV negative and HIV positive individuals as people living without HIV (PLWoH) and people living with HIV (PLWH) in all manuscript.
- As your comment, we changed words from HIV negative and HIV positive individuals to people living without HIV (PLWoH) and people living with HIV (PLWH) in all manuscript.
(Line 50-52)
“Although complete viral eradication and cure of HIV are not yet available, if HAART is started in a timely manner before immunodeficiency develops, life expectancy may be similar to that of people living without HIV.”
(Line 57-59)
“Tests available to monitor people living with HIV include HIV RNA, CD4+ T lymphocyte count, and complete blood count (CBC), of which the CD4+ T lymphocyte count is the most frequently used indicator in the evaluation of HIV infection.”
(Line 78-81)
“However, more than 3 million people living with HIVHIV infected patients reside within low to middle income countries where access to flow cytometry analyzers is limited, and it is in these developing countries where HIV infection is more prevalent.”
(Line 97-99)
“A total of 47 people living with HIV and 87 people living without HIV who visited Korea University Guro Hospital from September 2019 to February 2023 participated in this study.”
(Line 108-109)
“The FacsLyric was used as a flow cytometry for CD4 count in people living without HIV and people living with HIV patients.”
Comments 11: Introduction:
lines 82-83: cheaper and smaller than what?.
- As your comment, the sentence was changed followed as: (Line 86-87)
“The Microscanner Plus (Biozentech, Seoul, Republic of Korea) is a cell counter that is smaller, cheaper than a flow cytometer, and can be used without an expert.”
Comments 12: Materials and Methods
Lines: 144-145: it is not clear how the authors calculate the CD4 T cells count.
- As your comment, the sentence was changed followed as: (Line 163-168)
“The WBC number of each blood sample was measured using the Automated Hematology Analyzer DxH900 (Beckman Coulter, USA) in this study. CD4 and CD45 fractions of WBC were measured using both Microscanner Plus and FacsLyric flow cytometry, respectively. Based on this measured value, the number of CD4 + T lymphocytes can be calculated by dividing the CD4 fraction by the CD45 fraction and then multiplying by the WBC count.”
Reviewer 2 Report
Comments and Suggestions for Authors
The study conducted a comparison between the performance of Microscanner Plus and flow cytometry, which is meaning for HIV detection in resource-limited areas. However, the numbers of samples/patients are too less, I strongly suggest the authors to test more samples/patients, including both of HIV patients and healthy individuals. In addition, reproducibility of the Microscanner Plus is tested by two experimenters, which is also too less. I also suggest the authors to test the performance of Microscanner Plus by more experimenters.
Comments on the Quality of English Language1. Line 28-29, the numer 'six-hundred thousand deaths' is a yearly statistics or other time period? please be more clear.
2. Line 50, 'Although complete viral eradication and cure is yet available', not available or available, please double check.
3. Line 57, 'infection. [13-15].' should be 'infection [13-15].'
4. Line 65, the meaning of 'the diagnosis is made' is vague, please try to be more clear.
5. Line 77, 'results' should be 'result'.
6. Line 85, 'flowcytometry' should be 'flow cytometry'.
7. Line 113, 'censor' should be 'sensor'.
8. Line 153-154, 'For reproducibility, two independent experimenters each prepared same three samples and analyzed CD4 count of three samples' needs to be rephased.
9. Line 236-238. 'Common drawbacks of current methods are limited focus on mammalian cell observation, complexity of the interface, and the need for independent soft ware and computing resource-dependent image calculations.' needs to be rephased.
Author Response
Comments and Suggestions for Authors
The study conducted a comparison between the performance of Microscanner Plus and flow In response to your recommendation, we have expanded our study to include more participants. Specifically, we have added 10 additional HIV-positive patients and 50 HIV-negative individuals. With this expansion, the total number of participants in our study now includes 47 HIV-positive patients and 87 HIV-negative individuals. We have updated the related contents throughout the entire manuscript to reflect these changes.cytometry, which is meaning for HIV detection in resource-limited areas.
Comments 1: However, the numbers of samples/patients are too less, I strongly suggest the authors to test more samples/patients, including both of HIV patients and healthy individuals.
- Following your recommendation, we have expanded our study to include more participants. Specifically, we have added 10 additional HIV-positive patients and 50 HIV-negative individuals. With this expansion, the total number of 134 participants in our study now included 47 HIV-positive patients and 87 HIV-negative individuals. We have updated the related contents throughout the entire manuscript to reflect these changes.
Comments 2: In addition, reproducibility of the Microscanner Plus is tested by two experimenters, which is also too less. I also suggest the authors to test the performance of Microscanner Plus by more experimenters.
- As your comment, we conducted additional testing with four experimenters to assess the reproducibility of the Microscanner Plus. The content regarding the reproducibility has been updated as follows: (Line 209-230)
“In the reproducibility test, four experimenters, A, B, C, and D, performed measure-ments of CD4 counts. Experimenter A's measurements yielded mean CD4 counts of 1499.40, 740.17, and 314.57 cells/µL for the respective samples, with corresponding standard deviations (SD) of 11.90, 4.75, and 2.46 cells/µL. The coefficient of variation (CV) for these measurements was maintained below 1%, specifically at 0.79%, 0.64%, and 0.78%. Similarly, experimenter B reported mean CD4 counts of 1487.50, 746.13, and 314.20 cells/µL, with SDs of 20.61, 3.55, and 2.08 cells/µL, and CVs of 1.39%, 0.48%, and 0.66%. Experimenter C's results showed mean values of 1483.53, 741.77, and 317.73 cells/µL, with SDs of 24.77, 6.87, and 3.55 cells/µL, and CVs of 1.67%, 0.93%, and 1.12%. Lastly, experimenter D recorded means of 1495.43, 741.77, and 309.80 cells/µL, SDs of 13.74, 6.87, and 8.75 cells/µL, and CVs of 0.92%, 0.93%, and 2.82% (Table 2). All intra-individual CV values were within 5%, showing excellent reproducibility. Also, the ANOVA test revealed no statistically significant differences in CD4 counts among the four independent experimenters, as indicated by an F-statistic of 1.98 × 10-5.and a P value of 1.0.”
Comments on the Quality of English Language
Comments 1: Line 28-29, the numer 'six-hundred thousand deaths' is a yearly statistics or other time period? please be more clear.
- As your comment, we changed the sentence followed as: (Line 28-30)
“Human immunodeficiency virus (HIV) infection is responsible for over six-hundred thousand deaths world-wide in 2022 according to the World Health Organization (WHO) [1].”
Comments 2: Line 50, 'Although complete viral eradication and cure is yet available', not available or available, please double check.
- As your comment, we changed the sentence followed as: (Line 50-52)
“Although complete viral eradication and cure of HIV are not yet available, if HAART is started in a timely manner before immunodeficiency develops, life expectancy may be similar to that of people living without HIV.”
Comments 3: Line 57, 'infection. [13-15].' should be 'infection [13-15].'
- As your comment, we changed the sentence followed as: (Line 57-59)
“Tests available to monitor people living with HIV include HIV RNA, CD4+ T lymphocyte count, and complete blood count (CBC), of which the CD4+ T lymphocyte count is the most frequently used indicator in the evaluation of HIV infection [13-15].”
Comments 4: Line 65, the meaning of 'the diagnosis is made' is vague, please try to be more clear.
- As your comment, we changed the sentence followed as: (Line 66-68)
“Current treatment guidelines recommend that patients begin treatment as soon as possible once they are confirmed to be infected with HIV.”
Comments 5: Line 77, 'results' should be 'result'.
- As your comment, we changed the sentence followed as: (Line 81-82)
“Requirements of skilled operators, cost and resource dependent characteristics of flow cytometry result in limited access in such resource limited settings [19,20].”
Comments 6: Line 85, 'flowcytometry' should be 'flow cytometry'.
- As your comment, we changed the sentence followed as: (Line 89-91)
“To validate the performance of Microscanner Plus, a comparative evaluation was per-formed with flow cytometry as a standard method of CD4+ T lymphocyte counting.”
Comments 7: Line 113, 'censor' should be 'sensor'.
- As your comment, we changed the sentence followed as: (Line 121-122)
“Using a built-in CMOS-sensor camera and separate fluorescence light sources of Green 540 nm without emission filter, a Bright field (BF) image is obtained.”
Comments 8: Line 153-154, 'For reproducibility, two independent experimenters each prepared same three samples and analyzed CD4 count of three samples' needs to be rephased.
- As your comment, we changed the sentence followed as: (Line 173-177)
“For reproducibility, the CD4 count of each sample was analyzed by four independent re-searchers using the same three samples. Based on the CD4 counts analyzed by four independent experimenters, the CV value of the CD4 count between experimenters was calculated to confirm reproducibility.”
Comments 9: Line 236-238. 'Common drawbacks of current methods are limited focus on mammalian cell observation, complexity of the interface, and the need for independent soft ware and computing resource-dependent image calculations.' needs to be rephased.
- As your comment, we changed the sentence followed as: (Line 274-280)
“Common disadvantages of current methods include the complexity of the interface and the need for independent analysis software. Furthermore, it must be considered that most automated cell counters are limited in focus to only observe mammalian cells [31].”
Round 2
Reviewer 2 Report
Comments and Suggestions for Authors
For the future work, I suggest the authors can test more samples to prove their findings.